# Coping with Workplace Incivility in Hospital Teams: How Does Team Mindfulness Influence Prevention- and Promotion-Focused Emotional Coping?

**DOI:** 10.3390/ijerph192316209

**Published:** 2022-12-03

**Authors:** Samuel Farley, David Wei Wu, Lynda Jiwen Song, Rebecca Pieniazek, Kerrie Unsworth

**Affiliations:** 1Sheffield University Management School, University of Sheffield, Sheffield S10 1FL, UK; 2Business School, Renmin University of China, Beijing 100872, China; 3Leeds University Business School, University of Leeds, Leeds LS2 9JT, UK

**Keywords:** incivility, team mindfulness, coping

## Abstract

Incivility is a growing concern for researchers and practitioners alike, yet we know little about how the team context is related to the way that employees respond to it. In this study, we examined the role of team mindfulness and its direct and buffering effects on individual-level promotion- and prevention-focused emotional coping. We also examined how these forms of coping were related to individual work engagement. In a temporally lagged study of 73 hospital teams (involving 440 team members), multi-level analyses showed that team mindfulness was directly negatively associated with individual-level prevention-focused emotional coping (behavioral disengagement, denial, and venting); however, it was not positively related to individual-level promotion-focused forms of coping (positive reframing and acceptance). In addition, a cross-level interaction effect was identified whereby team mindfulness reduced the positive relationship between incivility and venting, meaning there was less individual-level venting following incivility in the context of higher team mindfulness. These findings may have implications for work engagement, which was shown to be negatively related to venting and behavioral disengagement. Our findings are useful for managers of teams that regularly experience customer incivility as it uncovers how they can develop a team context that discourages ineffective coping responses.

## 1. Introduction

Workplace incivility is an interpersonal stressor defined as low-intensity deviant behavior with ambiguous intent to harm [1], such as showing little interest in the opinions of others or addressing them in an unprofessional manner [2]. Despite being milder than bullying and social undermining, incivility has significant negative impacts on employee well-being [3,4], which can often be substantial [5]. Thus, it is important to understand how its negative consequences can be reduced.

One mitigating factor which has been considered in the literature is coping [6], which is defined as a person’s “constantly changing cognitive and behavioral efforts to manage specific external and/or internal demands that are appraised as taxing or exceeding the person’s resources” [7] (p. 993). Existing research has shed light on common coping responses to incivility [8], as well as the strategies that may have an adaptive and maladaptive impact [6,9]. Nevertheless, this research has primarily been conducted at the individual level, and thus we know little about how the team context influences the way that employees respond to incivility. This is surprising given that incivility occurs in a social context and hence would likely influence coping responses. Indeed, outside of the incivility context, recent research has found that team-level factors may significantly influence how employees appraise individual-level stressors [10], particularly interpersonal stressors [11]. 

In this paper, we examine the direct and moderating role of team mindfulness on individuals’ different coping responses to incivility. Specifically, we examine incivility experienced by hospital staff from a range of sources, including colleagues and patients. To the best of our knowledge, our study is the first to identify and evidence the role of a team-level factor in shaping how employees deal with incivility. By adopting social information processing theory (SIP) [12], we go beyond existing individual-level research [6,13] by showing how team mindfulness influences coping responses. 

### 1.1. Hypothesis Development

Researchers have studied various coping strategies that individuals adopt when exposed to incivility [6,8], which may broadly be categorized as problem-focused or emotion-focused [14]. Problem-focused coping involves taking action to change the negative situation, such as planning how to resolve it or actively taking steps to prevent it [14]. Alternatively, emotion-focused coping involves all regulative attempts to diminish the emotional consequences of stressful events [15]. Emotion-focused coping often occurs when individuals believe they cannot cope with the situation or have no means to change it [14]. 

In this study, we focus exclusively on emotion-focused coping strategies as prior research suggests that individuals most commonly adopt emotion-focused coping in response to incivility [16]. For example, one study found that 70% of incivility targets either ignored or accepted it [8], while another study found that targets ignored or avoided the perpetrator in over 70% of incivility incidents [17]. Cortina and colleagues [16] suggest emotion-focused coping occurs more commonly when people encounter incivility because taking steps to resolve the situation with the perpetrator can involve negatively charged interactions, which may threaten working relations and make interdependent working more challenging. This seemed to be the case in a study conducted by Hershcovis and colleagues [6], which found that confronting the perpetrator did not reduce subsequent experiences of incivility.

Although many forms of emotion-focused coping can be differentiated, we sought to investigate how incivility and team mindfulness were linked to promotion and prevention-based forms of emotional coping [18]. Promotion-focused coping involves “*efforts that maximize the chances for a match between one’s current situation and one’s hopes and aspirations*” [18] (p. 1297), and such strategies (e.g., acceptance, positive reframing) serve the goal of obtaining positive outcomes and an ideal state. In contrast, prevention-focused coping is defined as “*efforts that minimize the chances for a mismatch between one’s current situation and one’s duties and obligations*” [18] (p. 1297). Examples include venting, denial, and behavioral disengagement, which serve the strategy of preventing negative outcomes and failures. Since previous research finds that emotion-focused forms of coping are likely to be adopted more often that problem-focused forms of coping [16], we hypothesized that incivility would be positively associated with both promotion-focused and prevention-focused emotional coping:

**Hypothesis 1:** 
*Workplace incivility is positively associated with prevention-focused emotional coping in the form of denial, venting, and behavioral disengagement.*


**Hypothesis 2:** 
*Workplace incivility is positively associated with promotion-focused emotional coping in the form of acceptance and positive reframing.*


### 1.2. The Effects of Team Mindfulness

Team mindfulness refers to “a shared belief among team members that team interactions are characterized by awareness and attention to present events, and by experiential, non-judgmental processing of within-team experiences” [19] (p. 326). The construct involves two separate components: (1) present-focused attention, which involves “paying attention to what is happening in the moment”, and (2) experiential processing, which involves “observing stimuli in an open-minded fashion, without judgment or evaluation” [19] (p. 325–326). In this respect, team mindfulness is a shared perception regarding the extent to which team interactions are present-focused and non-evaluative. We argue that an attentive, open-minded, and non-evaluative team approach may buffer the effects of incivility and increase (or decrease) specific coping responses.

First, we suggest that team mindfulness will influence the relationship between incivility and coping outcomes. This is because SIP theory [12] argues that employees use the social information available to them to understand their work environments and to formulate attitudes and behaviors. Individual perceptions and actions are influenced by social contexts, especially when one is a part of a team [20]. Indeed, the mutual interdependence between team members creates a social context for them to have more interpersonal interactions, which means that team members can exert social influence over one another. In hospital settings, employees regularly work together in teams to provide patient care. However, the demanding nature of the medical context combined with needs of patients means that employees may regularly experience incivility from other hospital personnel or from patients and their families. 

According to SIP theory [12], the social context affects the saliency of information about employees’ activities and further helps team members to perceive information cues and behavioral responses [12]. Therefore, individuals who experience incivility or abusive supervision will use cues from their immediate social environment to appraise it and formulate an appropriate coping response [21]. For example, a nurse who experiences an uncivil comment from a patient may draw on her observations of how her colleagues have responded to the same patient to formulate her response. In such situations, the context of the hospital is somewhat unique as hospital personnel have caring responsibilities for their patients and may not have the same spectrum of coping options that are available when the perpetrator is an employee. Below, we consider how a mindful team context will influence individual-level coping responses. 

The nature of mindfulness involves noticing what is happening without making judgements about whether it is good or bad [22]. By actively processing an uncivil event, mindful team members are engaging directly with it, meaning they are less likely to adopt a preventive-focused emotional coping strategy such as denial, venting, or behavioral disengagement because such an approach would involve an appraisal of negativity. Moreover, the present-focused attention aspect of mindfulness suggests that team members may adopt acceptance when faced with an uncivil event. This is because acceptance involves acknowledging the stressor as real [23]. Mindful team members are also likely to adopt positive reframing when faced with incivility as the non-judgmental experiential processing element of mindfulness involves paying attention to events with curiosity, kindness, and compassion [19,24]. This suggests that mindful teams may process events in a manner that gives the instigator the benefit of the doubt. As a result, we argue that members of mindful teams will adopt promotion-focused emotional coping but not prevention-focused emotional coping in response to incivility. 

**Hypothesis 3:** 
*Team mindfulness is negatively related to prevention-focused emotional coping in the form of denial, venting, and behavioral disengagement.*


**Hypothesis 4:** 
*Team mindfulness is positively linked to promotion-focused emotional coping in the form of acceptance and positive reframing.*


**Hypothesis 5:** 
*Team mindfulness moderates the relationship between experienced incivility and prevention-focused emotional coping in the form of denial, venting, and behavioral disengagement, such that a more negative relationship will exist when mindfulness is higher.*


**Hypothesis 6:** 
*Team mindfulness moderates the relationship between experienced incivility and promotion-focused emotional coping in the form of acceptance and positive reframing, such that a more positive relationship will exist when mindfulness is higher.*


### 1.3. Coping and Work Engagement

To better understand the impact of different coping strategies on employee outcomes, we sought to examine how they related to work engagement, defined as “*a positive, fulfilling, work related state of mind that is characterized by vigor, dedication, and absorption*” [25] (p. 74). A meta-analysis of coping efficacy found that promotion-focused emotional coping was positively related to employee outcomes such as performance, attitudes, and well-being [18]. In contrast, prevention-focused emotional coping was negatively related to these outcomes. When focusing on employee attitudes, Zhang and colleagues [18] suggested that promotion-focused emotional coping increased peoples’ sensitivity to positive environmental cues about their job whereas prevention-focused emotional coping increased peoples’ attention on the negative aspects of their jobs. This in turn influences how engaged and committed people are to their work. 

**Hypothesis 7:** 
*Prevention-focused emotional coping in the form of denial, venting, and behavioral disengagement is negatively related to work engagement.*


**Hypothesis 8:** 
*Promotion-focused emotional coping in the form of acceptance and positive reframing is positively related to work engagement.*


## 2. Materials and Methods

### 2.1. Participants and Procedure

Data were collected from 526 employees of a large Chinese hospital who were organized into 82 teams (average team size was 6.41). This context was chosen as medical staff are often exposed to incivility from patients and other organizational outsiders [26]. Their mean age was 31.5 (SD = 7.5); 388 were female (73.2%), 54 were male (10.2%), and 88 (16.6%) did not state their gender. Most participants worked as nurses (*n* = 411, 78.1%), and the others worked as administrators (*n* = 79, 14.3%) or pharmacists (*n* = 40, 7.6%). The nurse teams were generally organized according to medical disciplines, where a head nurse was responsible for the other nurses within his/her team. The pharmacists were also organized into teams based on discipline, while the administration teams were organized based on their department (i.e., each department acted as a team).

Survey data were collected at three time points separated by one month. This response period was adopted as prior research has shown that a shorter assessment period is required to capture meaningful relationships between incivility and outcome variables [27]. After accounting for attrition across the three time points, the final sample included 440 individuals within 73 teams. In the first survey, employees were asked about their age, gender, and incivility experiences. In the second survey, they were asked about team mindfulness. In the third survey, they were asked about their work engagement and the strategies they used to cope with incivility, including the extent to which they engaged in acceptance, positive reframing, denial, venting, and behavioral disengagement. The main advantage of capturing team mindfulness at the second time point was to limit common method variance. 

### 2.2. Measures

All surveys were in Chinese; we used translation and back-translation to ensure language equivalence. Unless indicated otherwise, all items were rated on a seven-point Likert scale ranging from 1 = strongly disagree to 7 = strongly agree.

Incivility was measured using Cortina and colleagues’ seven-item scale [2], which asked about employee experiences of incivility over the past month. The scale reliability was 0.91, and an example item is ‘*ignored or excluded you from professional camaraderie’*. We also asked respondents to specify who was the main perpetrator of the incivility and provided the following response options: ‘managers’, ‘consultants’, ‘doctors’, ‘nurses’, ‘patients’, ‘relatives of patients’, ‘other’, and ‘I did not experience any behaviors’.

Team mindfulness was measured using a ten-item scale developed by Yu and Zellmer-Bruhn [19]. The aggregation at team level fitted adequately (rwg = 0.79, ICC(1) = 0.11, ICC(2) = 0.42), and the scale reliability was 0.87. A sample item is *“The team is aware of thoughts and feelings without over-identifying with them.”*

Promotion-focused emotional coping was measured through two indicators: acceptance and positive reframing. Both were measured using two-item scales developed by Carver [28]. The scale reliability of the positive reframing scale was 0.90, and a sample item is ‘*I have been trying to see it in a different light to make it seem more positive*’. The alpha (α) of the acceptance scale was 0.83, and a sample item is ‘*I have been learning to live with it’*. Three forms of prevention-focused emotional coping were assessed, including denial (α = 0.92), venting (α = 0.88), and behavioral disengagement (α = 0.92). All three were assessed using two-item scales developed by Carver [28]. A sample item from the denial scale is *‘I have been saying to myself this isn’t real’*. A sample item from the venting scale is ‘*I have been expressing my negative feelings*’. A sample item from the behavior disengagement scale is ‘*I have been giving up trying to deal with it’*. 

Work engagement was measured by the UWES-9 scale developed by Schaufeli et al. [29]. The measurement consists of three subdimensions, including vigor, dedication, and absorption. The reliability was 0.95. A sample item is *“At my work, I feel that I am bursting with energy.”*

We controlled for age, gender, and team size in the analyses. We controlled for age as it has been shown to affect how people cope with mistreatment [30]. We controlled for gender as females were over-represented in the sample. Finally, we controlled for team size as larger teams experience more conflict [19], which may be experienced as incivility. 

### 2.3. Data Analysis

Multi-level modelling was undertaken to test hypotheses 1–8 using Mplus version 8.5 (see Figure 1). This allowed us to understand how our level 2 predictor (team mindfulness) influenced the level 1 coping outcomes (cross-level direct effects), as well as how it influenced the relationship between incivility and coping at the lower level (cross-level interaction effects). ICC(1) values showed that between 5.3% and 19.3% of the variance in the coping variables was attributable to the team level, which confirms the appropriateness of multi-level analyses (see Table 1 for correlation matrix). To test the hypotheses, we sequentially fitted a series of more complex models for each coping outcome. We first examined the baseline (null) model, which provides information on the proportion of variance in the coping outcome that is attributable to the individual and team levels. We then examined the fixed effects of the control variables on the coping outcome, followed by the fixed effects of incivility (controlling for age, gender, and team size). Fixed effects models examine whether a relationship is constant across all level 2 units (in this case, teams). In the subsequent model, we examined the fixed effects of team mindfulness on the coping outcome (with incivility and the control variables in the model). We then assessed the random effects of incivility, whereby the impact of incivility on each coping outcome was allowed to vary across the teams. Finally, we tested for cross-level interactions by examining whether team mindfulness influenced the relationship between incivility and coping. The models were tested with a maximum likelihood estimator, and we used full information likelihood estimation to deal with missing data. For the full model results of each coping outcome, please see the Appendix A.

## 3. Results

Prior to the formal analyses, we explored the data to better understand the nature of the incivility experienced by the respondents. At time 1, 474 respondents (90.1%) had experienced some form of incivility in the past month. However, of these only 157 (33.1%) were willing to indicate the source of the incivility. These 157 respondents most commonly cited patients’ relatives as the main perpetrators (55.4% of cases), followed by patients (24.2%), nurses (7.6%), managers (6.4%), doctors (4.5%), and consultants (1.9%). 

### 3.1. Hypotheses 1 and 2 

To test Hypotheses 1 and 2, we fitted models that examined the fixed effects of incivility on the coping outcomes. A fixed effects model was adopted as we expected the relationship between incivility and coping to be consistent across all team members in the sample. Age, gender, and incivility were specified as within-level variables, whereas team size was specified as a between-level variable. Age and incivility were both group-mean centered. Hypothesis 1 was supported as the fixed effects of incivility on prevention-focused emotional coping were positive and significant for denial (unstandardized B coefficient (B) = 0.17, *p* < 0.01), venting (B = 0.20, *p* < 0.01), and behavioral disengagement (B = 0.27, *p* < 0.001). However, Hypothesis 2 was not supported as incivility was unrelated to positive reframing (B = −0.01, *p* = 0.88) and acceptance (B = 0.02, *p* = 0.69). 

### 3.2. Hypotheses 3 and 4

To test Hypotheses 3 and 4, we fitted models that examined the fixed effects of team mindfulness on the coping variables. A fixed effects model was again adopted as we expected the hypotheses to be consistent for all teams in the sample. Age, gender, and incivility were assigned as within-level variables, whereas team size and team mindfulness were specified at the between-level. Age and incivility were again group-mean centered. Hypothesis 3 was fully supported as team mindfulness was negatively related to prevention-focused emotional coping in the form of denial (B = −0.46, *p* < 0.05), venting (−0.74, *p* < 0.001), and behavioral disengagement (B = −0.53, *p* < 0.05). However, Hypothesis 4 was unsupported as team mindfulness was not significantly related to positive reframing (B = −0.20, *p* = 0.21) or acceptance (B = −0.19, *p* = 0.20). These results suggest that team mindfulness influences prevention-focused emotional coping over and above the impact of incivility, but not promotion-focused emotional coping. 

### 3.3. Hypotheses 5 and 6

To test Hypotheses 5 and 6, we allowed the relationship between incivility and the coping outcomes to vary across the teams within our sample (a random effects model). We then sought to determine the extent to which team mindfulness explained variation in the incivility–coping relationships by testing for cross-level interactions. Hypothesis 5 was only partially supported as team mindfulness moderated the individual-level relationship between incivility and venting (B = −0.47, *p* < 0.05), such that at high levels of team mindfulness the relationship is more negative. This indicates that team mindfulness limits the extent to which team members engage in venting when they experience incivility. However, team mindfulness did not significantly influence the relationships between incivility, denial (B = −0.25, *p* = 0.25), and behavioral disengagement (B = −0.23, *p* = 0.26). Hypothesis 6 was not supported as team mindfulness did not moderate the individual-level relationship between incivility and positive reframing (B = −0.04, *p* = 0.79) or between incivility and acceptance (B = −0.15, *p* = 0.32). 

### 3.4. Hypotheses 7 and 8

Finally, to test Hypotheses 7 and 8 we fitted two regression models whereby work engagement was regressed on the promotion-focused and prevention-focused emotional coping variables separately. In these models, we controlled for age and gender, which were shown to be significantly related to work engagement. Hypothesis 7 was partially supported as venting (B = −0.15, *p* < 0.01) and behavioral disengagement (B = −0.15, *p* < 0.05) were significantly negatively related to work engagement but denial (B = 0.10, *p* = 0.11) was not. Hypothesis 8 was partially supported as positive reframing was positively related to work engagement (B = 0.23, *p* < 0.001) but acceptance (B = 0.00, *p* <0.99) was not.

## 4. Discussion

Across a sample of 440 employees who were organized into 73 teams, our findings suggest that those who regularly encounter incivility are most likely to respond with prevention-focused emotional coping strategies, including denial, venting, and behavioral disengagement. However, we extend previous research on coping responses at the individual level [6,8] as our results suggest that team mindfulness mitigates these effects by buffering incivility’s effect on venting and by limiting denial, venting, and behavioral disengagement. This may have downstream implications for employee work engagement, which was negatively related to two of the three prevention-focused emotional coping strategies. 

To our knowledge, this is the first study that has explicitly examined how team mindfulness influences coping responses to incivility, and it is notable that it was negatively associated with prevention-focused responses but not positively related to promotion-focused strategies. Mindfulness is growing in popularity across a range of organizations; indeed, Microsoft recently integrated the Headspace app onto its platform to encourage people to meditate while working on the computer. To support this development, the accompanying body of literature to assist its use is also growing; however, much of this literature is at the individual level [31,32]. Although our understanding of team mindfulness is advancing [33,34], our current knowledge is based on its more general effects rather than its role in stressful situations [34,35,36]. To our knowledge, this is the first study that has explicitly examined how team mindfulness influences coping responses to incivility.

Given that mindfulness involves the non-judgmental appraisal of events, it is unsurprising that employees within this team environment do not respond to incivility by venting. However, it is not clear why a similar finding was not observed in relation to denial and behavioral disengagement, both of which entail some form of negative appraisal. One potential explanation is that the hospital context meant that participants were unable to engage in such withdrawal-type behaviors regardless of team mindfulness as all hospital employees faced with incivility need to continue interacting with their patients to some degree, regardless of their team context. Venting, on the other hand, is distinct from the patients and therefore may be more affected by the team context.

Contrary to our expectations, mindfulness was unrelated to the promotion-focused emotional coping strategies of acceptance and positive reframing. One explanation for this non-significant finding is that at its core, team mindfulness is a “*shared cognitive state in which team members’ interactions are typified by attention to and non-judgmental processing of present events*” [33] (p. 432). By observing and being non-judgmental about different events, team mindfulness discourages negative appraisals but it does not necessarily encourage acceptance or positive reframing of the situation. This subtle distinction between non-judgmental observation and non-judgmental acceptance may reflect a difference between how mindfulness is characterized at the individual and team levels. Acceptance is traditionally an aspect of how mindfulness is conceptualized as the individual level [37]. However, acceptance cannot occur in the same way at the team level without debate and judgement among team members. 

When examining how the coping responses were related to work engagement, we found partial support for our hypotheses. Of the prevention-focused strategies, venting and behavioral disengagement were significantly negatively related to work engagement but denial was not. Similarly, the promotion-focused strategy of positive reframing was significantly positively associated with work engagement but acceptance was not. Although this pattern of results reflected our expectations and is in line with previous research [18], further research is required to determine why some of the coping strategies were unrelated to work engagement. It is possible that denial and acceptance reflect a more passive way of coping with incivility, which would not affect individuals’ work engagement. 

Our findings are nonetheless helpful for employees struggling with adverse impacts of experiencing incivility. Given that we found that a context of team mindfulness can both directly and indirectly reduce the enactment of the less desirable forms of coping with incivility, organizations where staff experience hostility from clients/customers should attempt to increase team mindfulness. However, given that the phenomenon of team mindfulness is in its infancy, we should not assume that training designed to increase mindfulness at the individual level [38] will necessarily be appropriate for increasing mindfulness at the team level. Hence, we recommend that future researchers aim to design and demonstrate the effectiveness of team mindfulness interventions, especially given that current mindfulness training delivered in workplaces is thought to typically differ from the training protocols recommended by scientific research [31].

## 5. Limitations, Future Directions, and Conclusions

Some limitations should be acknowledged. First, although we used a time-separated design that limits the likelihood of common method bias, we cannot infer causality, and therefore the possibility of alternative explanations for our findings should be considered. For example, it may be the case that coping responses shape team climates rather than vice versa. Second, we did not examine whether coping responses differed according to the source of the incivility, which may be an important moderator. For example, Hershcovis and colleagues [39] found that the perpetrator’s power and task interdependence can affect how targets of aggression respond. Therefore, future studies should examine how the source of the incivility influences the coping response, especially when the source may be part of the team context. Finally, although we only examined one team-level variable in this study, the findings suggest that team climates and emergent states may play an important role in how individuals cope with workplace aggression. We therefore encourage researchers to examine other team-level variables such as team servant leadership [40], which may buffer the negative impact of workplace mistreatment.

Organizations are starting to adopt mindful practices to improve the well-being, performance, and relationships of employees. Our study contributes to a growing area of research on team mindfulness by showing that it influences how employees cope with incivility. Most notably, members of mindful teams engaged in less venting when they experienced incivility. Although venting may be a useful strategy in some contexts, we found that it was negatively related to work engagement. Therefore, whilst further research is needed, our study tentatively demonstrates the benefits of a mindful team context. We also contribute more broadly to the research on how incivility affects coping outcomes by showing that those who encounter uncivil actions tend to adopt prevention-focused emotional coping strategies rather than promotion-focused ones. 

## Figures and Tables

**Figure 1 ijerph-19-16209-f001:**
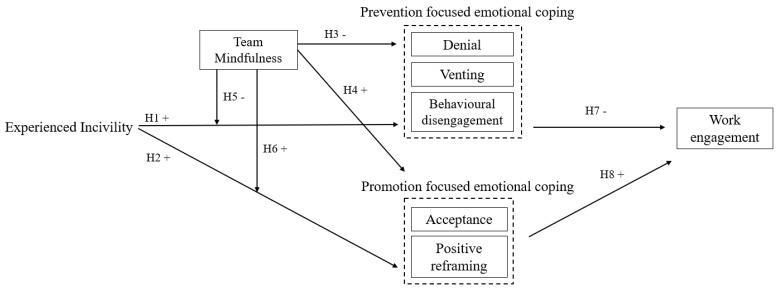
Hypotheses tested in the study.

**Table 1 ijerph-19-16209-t001:** Descriptive statistics and correlations.

Variables	M	SD	1	2	3	4	5	6	7	8	9	10
1 Age	31.49	7.45										
2 Gender (M = 0, F = 1)	0.88	0.33	−0.21 **									
3 Incivility	2.69	1.13	0.00	−0.01								
4 Acceptance	5.26	1.02	−0.04	−0.04	0.01							
5 Positive reframing	5.38	0.95	0.07	−0.04	0.02	0.54 **						
6 Denial	3.34	1.41	0.12 *	−0.20 **	0.13 **	0.08	0.04					
7 Venting	3.35	1.43	0.03	0.09	0.15 **	0.05	0.03	0.36 **				
8 Behav. Disengage.	3.06	1.36	0.04	−0.04	0.21 **	−0.03	−0.09	0.64 **	0.52 **			
9 Work engagement	5.28	1.02	0.33 **	−0.20 **	−0.06	0.11 *	0.24 **	0.00	−0.20 **	−0.15 **		
*10 Team size*	*6.41*	*3.56*	*−0.11*	*0.26 **	*−0.09*	*0.24 **	*0.20*	*−0.10*	*0.07*	*−0.12*	*−0.04*	
*11 Team mindfulness*	*4.63*	*0.42*	*0.18*	*−0.05*	*−0.49 ***	*−0.28 **	*−0.16*	*−0.21*	*−0.32 ***	*−0.23*	*0.18*	*0.00*

Nperson = 440, NTeam = 73; Level 2 variables in italics; Level 1 variables were aggregated to provide estimates of between-team relationships with level 2 variables. ** Significant at 0.01 level; * Significant at 0.05 level.

## Data Availability

Please contact the first author for further details.

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
