# Peer review of "Coping with Workplace Incivility in Hospital Teams: How Does Team Mindfulness Influence Prevention- and Promotion-Focused Emotional Coping?"

_ijerph, 2022, doi:10.3390/ijerph192316209_

Round 1

Reviewer 1 Report

Hello - thanks for the opportunity to review this article. It represents an interesting study on a very important topic - consideration of the issues of "team mindfulness " and " Coping with Workplace Incivility". 

 Please see the detailed comments below for specific issues:

Abstract: abstract is clear.

Introduction:

From the beginning of the introduction I have problems to understand the origin of "Workplace incivility" handled in the research. Although later, in the limitations of study it appears somewhat nuanced, I would like it to be better defined in the introduction whether reference is being made to "Workplace incivility" within a team, between other colleagues outside the team or between patient and health care personnel. 

In the introduction, the importance of analyzing the social context from the SIP theory (line 99) appears repeatedly.  However, nowhere does it appear that the study was carried out in a hospital. This workplace seems to me to be a significant context with respect to others and should be briefly analyzed in the introduction, even if it is mentioned in the title. We cannot consider that the Hospital is an interesting setting simply because they are often exposed to incivility and we can change it for another one.

I do not understand what the section "Coping and Work Engagement" (line 139) contributes to the research.  The hypotheses derived from this section are not presented in the discussion section.

. Materials and Methods

The sample presents a gender imbalance (line 162). In this sense, I think that a cross-sectional gender view of the work is interesting. Could the research have a gender bias that explains the data?

Why are there no doctor included in the sample?

I have some doubts about the teams: What are the composition criteria?, Are they interdisciplinary teams? Are the teams organized hierarchically? etc.

Results

The gender analysis is confusing.  Perhaps a Descriptive Statistics analysis by gender could be clarifying. Likewise, in the analysis of partial correlations should have gender controlled.

Dicussion

The hypotheses derived from the section "Coping and Work Engagement" are not presented in the discussion section.

Author Response

Dear Ms Yang,

Thank you for the opportunity to revise manuscript ijerph-2021717, now titled “Coping with workplace incivility in hospital teams: How does team mindfulness influence prevention and promotion focused emotional coping”, for further consideration at IJERPH. Thank you also for the helpful and constructive feedback on our original submission and for that of the reviewers.

Below, we outline our responses to each of the specific comments made. For ease of viewing, we have included the original comments in bold text and our own responses in regular text. The changes we have made to the manuscript appear in red type in the manuscript itself, and in the letter we include the page references and any quoted text is italicised.

We believe that the changes we have made to our manuscript, in response to these thoughtful comments, have produced a clearer contribution to the literature. We hope you will agree that in making these changes the manuscript is much improved.

Thank you once again for the opportunity to revise this work.

Sincerely,

Authors

Reviewer 1 comments and responses

Hello - thanks for the opportunity to review this article. It represents an interesting study on a very important topic - consideration of the issues of "team mindfulness " and " Coping with Workplace Incivility". 

Thank you very much for your interest in our work and for your positive feedback. Thank you also for the effort invested into the review process and your constructive advice helping us to improve the manuscript.  

 Please see the detailed comments below for specific issues:

Abstract: abstract is clear.

Introduction:

From the beginning of the introduction I have problems to understand the origin of "Workplace incivility" handled in the research. Although later, in the limitations of study it appears somewhat nuanced, I would like it to be better defined in the introduction whether reference is being made to "Workplace incivility" within a team, between other colleagues outside the team or between patient and health care personnel.

Thank you for this comment. In the introduction, we have made clear on page 2 that we examined incivility enacted by both colleagues (within and outside the team), as well as patients and their families. Moreover, in the method section, on page 4 we note that we also recorded the main source of the incivility by asking participants to state the main perpetrator using the response scale: managers’, ‘consultants’, ‘doctors’, ‘nurses’, ‘patients’, ‘relatives of patients’, ‘other’, and ‘I did not experience any behaviors’. The results from this item are now presented in the results on page 6, specifically we write:

Prior to the formal analyses, we explored the data to better understand the nature of the incivility experienced by the respondents. At time 1, 474 respondents (90.1%) had experienced some form of incivility in the past month. However, of these, only 157 (33.1%) were willing to indicate the source of the incivility. These 157 respondents most commonly cited patients’ relatives as the main perpetrators (55.4% of cases), followed by patients (24.2%), nurses (7.6%), managers (6.4%), doctors (4.5%), and consultants (1.9%).        

In the introduction, the importance of analyzing the social context from the SIP theory (line 99) appears repeatedly.  However, nowhere does it appear that the study was carried out in a hospital. This workplace seems to me to be a significant context with respect to others and should be briefly analyzed in the introduction, even if it is mentioned in the title. We cannot consider that the Hospital is an interesting setting simply because they are often exposed to incivility and we can change it for another one.

We agree that the hospital context should be made clearer. We have now amended the title to indicate the setting of the study: ‘Coping with workplace incivility in hospital teams: How does team mindfulness influence prevention and promotion focused emotional coping?’. We also state in the introduction on page 2 that we examine the incivility experienced by hospital personnel.

I do not understand what the section "Coping and Work Engagement" (line 139) contributes to the research.  The hypotheses derived from this section are not presented in the discussion section.

The section on coping and work engagement acts as a validation check on the research. In the paper, we argue that the prevention focused emotional coping strategies tend to be undesirable, while promotion focused emotional coping methods tend to be more beneficial. By examining how the coping variables relate to an outcome such as work engagement, it is possible to provide some tentative empirical evidence for this viewpoint. Nevertheless, you are right to point out that we should address the findings from these hypotheses in the discussion. We have therefore included the following paragraph in the discussion section, which addresses these hypotheses directly:

“When examining how the coping responses were related to work-engagement, we found partial support for our hypotheses. Of the prevention focused strategies, venting and behavioral disengagement were significantly negatively related to work engagement, but denial was not. Similarly, the promotion focused strategy of positive reframing was significantly positively associated with work engagement, but acceptance was not. Although this pattern of results reflected our expectations and is in line with previous research [18], further research is required to determine why some of the coping strategies were unrelated to work engagement. It is possible that denial and acceptance reflect a more passive way of coping with incivility, which would not affect individuals’ work engagement.”   

Materials and Methods

The sample presents a gender imbalance (line 162). In this sense, I think that a cross-sectional gender view of the work is interesting. Could the research have a gender bias that explains the data?

Although far more females were included in the research than males, our results do not suggest that gender unduly affected the findings. First, as noted on page 5, we controlled for gender in all statistical tests, therefore any impact of gender was held constant and would not have affected the findings. Second, the correlation table shows that the relationship between incivility and gender was 0, and apart from denial (which was used slightly more by female participants .12*, p <.05), there was no difference in the coping strategies used by male and female respondents.  

Why are there no doctor included in the sample?

We attempted to include doctors in the sample, but the hospital where we collected data did not agree to this, presumably as their time needed to be prioritised for patient care.

I have some doubts about the teams: What are the composition criteria?, Are they interdisciplinary teams? Are the teams organized hierarchically? etc.

Most teams were organized hierarchically. All teams were divided by their internal reporting relationship, namely a line manager and his/her subordinates. The nurse team were generally based on medical disciplines, and there was a head nurse for each discipline, with several nurses who reported to the head nurse. The pharmacists were also organised into teams based on discipline, while the administration teams were organised based on their department (i.e. each department acted as a team).

Results

The gender analysis is confusing.  Perhaps a Descriptive Statistics analysis by gender could be clarifying. Likewise, in the analysis of partial correlations should have gender controlled.

As gender was presented in the correlation table, we are not sure how presenting descriptive statistics by gender would benefit the manuscript. As noted above, the correlation table shows that gender was positively related to denial but was otherwise unrelated to all other coping variables and incivility. Indeed, of all the variables included in the study, gender was only significantly linked to age, denial, and work engagement, and the correlation table shows the direction of these relationships. Since we also controlled for gender in all the analyses, we do not feel that presenting descriptive statistics by gender would be worthy of the readers’ attention. 

Discussion

The hypotheses derived from the section "Coping and Work Engagement" are not presented in the discussion section.

As noted above, we have now included a paragraph to address these findings in the discussion.

Thank you again for your advice and suggestions, which we believe have helped us to clarify and strengthen the contribution of this work.

Reviewer 2 Report

Dear Authors, I am happy to review this paper. I must say that the research has been conceptualized well, and the methods part is good. However, the results section is sketchy and the discussion looked incoherent. Similarly, the conclusion could have been better. Let me give some specific examples: 

1. Why different variables have been measured at 3-time intervals? The idea is fine, but there is no justification for this. It should have explained why mindfulness has been assessed at time 2. 

2. There should have been a conceptual model connecting various variables to get better clarity. Please add this.

3.  Data Analysis section could have been more elaborate. E.g., what is a baseline model? What are fixed effects and random effects? What specific analyses have been used to assess these?

4. "To test the hypotheses, we sequentially fitted a series of more complex models for each coping outcome" has been written by the authors in the manuscript, surprisingly,  there is just one Correlation Table. The authors should have provided some more Tables of results for better understanding. 

5. Some of the results (beta values, and their interpretation) appear to be reported wrongly. At some places, the Beta value of .25 has been reported to be insignificant while at other places .15 has been reported as significant. Please check carefully. 

6. Discussion generally follows a systematic pattern, the first paragraph briefly presents the aim of the research, then all the hypotheses are presented one by one and discussed. Lastly, the discussion is summarized at the end. This was missing.

7. Conclusion focused only on mindfulness and did not speak anything about other important aspects of the work. Please expand to include all your major findings. 

Author Response

Dear Ms Yang,

Thank you for the opportunity to revise manuscript ijerph-2021717, now titled “Coping with workplace incivility in hospital teams: How does team mindfulness influence prevention and promotion focused emotional coping”, for further consideration at IJERPH. Thank you also for the helpful and constructive feedback on our original submission and for that of the reviewers.

Below, we outline our responses to each of the specific comments made. For ease of viewing, we have included the original comments in bold text and our own responses in regular text. The changes we have made to the manuscript appear in red type in the manuscript itself, and in the letter we include the page references and any quoted text is italicised.

We believe that the changes we have made to our manuscript, in response to these thoughtful comments, have produced a clearer contribution to the literature. We hope you will agree that in making these changes the manuscript is much improved.

Thank you once again for the opportunity to revise this work.

Sincerely,

Authors

Reviewer 2 comments and responses

Dear Authors, I am happy to review this paper. I must say that the research has been conceptualized well, and the methods part is good. However, the results section is sketchy and the discussion looked incoherent. Similarly, the conclusion could have been better. Let me give some specific examples: 

Thank you for your interest and feedback. We hope that the revised paper addresses your concerns.

  1. Why different variables have been measured at 3-time intervals? The idea is fine, but there is no justification for this. It should have explained why mindfulness has been assessed at time 2. 

Thank you for this point. We have included the rationale for the response window in the revised manuscript on page 4. Specifically we write:

“Survey data were collected at three time points, separated by a month. This response period was adopted as prior research has shown that a shorter assessment period is required to capture meaningful relationships between incivility and outcome variables [27]”

This justification is from a paper by Matthews and Ritter (2016), which indicates that asking participants to recall their experiences of incivility over a longer period may introduce ‘noise’ and be distorted by memory bias. The main benefit of capturing team mindfulness at a separate time point is to limit the likelihood of common method variance, which we feel is an advantage of our research design.

  1. There should have been a conceptual model connecting various variables to get better clarity. Please add this.

Thank you, we agree that a conceptual model would add clarity. We have added this figure on page 5.  

  1. Data Analysis section could have been more elaborate. E.g., what is a baseline model? What are fixed effects and random effects? What specific analyses have been used to assess these?

We agree that more detail needed to be provided in the results section. We have therefore added information to explain the model testing procedure in more detail, along with a clearer explanation of fixed and random effects. Specifically, on page 5, we note:

“Multi-level modelling was undertaken to test hypotheses 1-8 using Mplus version 8.5 (see figure 1). This allowed us to understand how our level 2 predictor (team mindfulness) influenced the level 1 coping outcomes (cross-level direct effects), as well as how it influenced the relationship between incivility and coping at the lower level (cross-level interaction effects). ICC(1) values showed that between 7.2% and 19.4% of the variance in the coping variables was attributable to the team-level, which confirms the appropriateness of multi-level analyses (see Table 1 for correlation matrix). To test the hypotheses, we sequentially fitted a series of more complex models for each coping outcome. We first examined the baseline (null) model, which provides information on the proportion of variance in the coping outcome that is attributable to the individual and team level. We then examined the fixed effects of the control variables on the coping outcome, followed by the fixed effect of incivility (controlling for age, gender, and team size). Fixed effect models examine whether a relationship is constant across all level 2 units (in this case teams). In the subsequent model, we examined the fixed effect of team mindfulness on the coping outcome (with incivility and the control variables in the model). We then assessed the random effect of incivility, whereby the impact of incivility on each coping outcome was allowed to vary across the teams. Finally, we tested for cross-level interactions by examining whether team mindfulness influenced the relationship between incivility and coping. The models were tested with a maximum likelihood estimator and we used full information likelihood estimation to deal with missing data. For the full model results of each coping outcome, please contact the first author.”  

  1. "To test the hypotheses, we sequentially fitted a series of more complex models for each coping outcome" has been written by the authors in the manuscript, surprisingly, there is just one Correlation Table. The authors should have provided some more Tables of results for better understanding. 

You are right to point out that we generated many statistical models in the analyses. For example, for each coping outcome we fitted a series of six different models (30 in total). In the interests of brevity, we have stated that readers are able to contact the first author for the full model results if they wish (p. 5). Therefore, interested readers can obtain further results if they wish, but at the same time our manuscript is not overloaded with tables.  

  1. Some of the results (beta values, and their interpretation) appear to be reported wrongly. At some places, the Beta value of .25 has been reported to be insignificant while at other places .15 has been reported as significant. Please check carefully. 

Multilevel modelling does not often use standardized coefficients, so all of the B values reported in the paper are the unstandardized B coefficients (rather than beta values - we have now made this clear on page 6 of the paper). The unstandardized B coefficient is not comparable across different variables. It would therefore not be surprising if a B of .15 was significant for a relationship involving one set of variables, but not another (particularly if the latter relationship involved a team level variable).

  1. Discussion generally follows a systematic pattern, the first paragraph briefly presents the aim of the research, then all the hypotheses are presented one by one and discussed. Lastly, the discussion is summarized at the end. This was missing.

We have now taken steps to ensure that the results from all hypotheses were discussed in the manuscript, as we have added a paragraph on page 8 in which we review hypotheses 7 and 8. Although we agree that it is important to address all the findings within the discussion, we sought to structure it by drawing out the main theoretical contributions of the research and what they mean in practice, as opposed to discussing each hypothesis in turn. In doing so, we sought to avoid one of the main pitfalls to writing discussion sections outlined by Geletkanycz and Tepper (2012, p. 258):

“A common mistake authors make is to devote too much discussion to summarizing and resummarizing the results of their hypothesis tests while devoting too little attention to explaining what the results mean. In some cases, authors restate the findings in the first few paragraphs of the Discussion section and then move on to other subsections (practical implications, limitations, future research directions, and so on) without addressing the study’s theoretical implications whatsoever”.

Geletkanycz, M., & Tepper, B. J. (2012). Publishing in AMJ–part 6: Discussing the implications. Academy of management journal, 55(2), 256-260.

  1. Conclusion focused only on mindfulness and did not speak anything about other important aspects of the work. Please expand to include all your major findings. 

We have expanded this section on page 10 to better highlight the main findings of the research.

Thank you again for your advice and suggestions, which we believe have helped us to clarify and strengthen the contribution of this work.

Reviewer 3 Report

Human relationships in the workplace are very subtle and especially difficult to solve because they are personal problems. In addition, it is very difficult to derive a general method for solving human relationships in the workplace. From this point of view, this study can be said to be a very rare and valuable study that approached human relations in the workplace from a Team Mindfulness perspective from various perspectives such as cause and prevention.

In order to improve the completeness of this paper, please revise the following.
- Since the people who participated in the survey are those who work in Chinese hospitals, it would be better to write "Focused on Hospital Works" as an example in the title of the paper.
-Team Mindfulness is the most important keyword of this study. In Section 1.2, it is necessary to describe or define their definition of Team Mindfulness by the authors of this paper.
-The questionnaire is written in Chinese. Is it sure that there are no errors in the translation process into English? This is because the questionnaire itself is bound to be somewhat abstract.
- On the other hand, describe whether it is confident that the 526 participants in the survey gave very true answers.
-Chapter 4. Discussion has very abstract contents, so it is recommended to present the overall argument as an easily understandable picture.
-4.1 Limitations and Future Directions should be combined with Chapter 5. In other words, it is recommended to change the title of Chapter 5 to 5. Conclusions and Future Research Works and describe future research tasks. On the other hand, the limitations of this study described in Section 4.1 are natural due to the nature of this study, so there is no need to describe it.

Author Response

Dear Ms Yang,

Thank you for the opportunity to revise manuscript ijerph-2021717, now titled “Coping with workplace incivility in hospital teams: How does team mindfulness influence prevention and promotion focused emotional coping”, for further consideration at IJERPH. Thank you also for the helpful and constructive feedback on our original submission and for that of the reviewers.

Below, we outline our responses to each of the specific comments made. For ease of viewing, we have included the original comments in bold text and our own responses in regular text. The changes we have made to the manuscript appear in red type in the manuscript itself, and in the letter we include the page references and any quoted text is italicised.

We believe that the changes we have made to our manuscript, in response to these thoughtful comments, have produced a clearer contribution to the literature. We hope you will agree that in making these changes the manuscript is much improved.

Thank you once again for the opportunity to revise this work.

Sincerely,

Authors

Reviewer 3

Human relationships in the workplace are very subtle and especially difficult to solve because they are personal problems. In addition, it is very difficult to derive a general method for solving human relationships in the workplace. From this point of view, this study can be said to be a very rare and valuable study that approached human relations in the workplace from a Team Mindfulness perspective from various perspectives such as cause and prevention.

Thank you very much for your interest in our work and for your positive feedback. Thank you also for the effort invested into the review process and your constructive advice helping us to improve the manuscript.  

In order to improve the completeness of this paper, please revise the following.
- Since the people who participated in the survey are those who work in Chinese hospitals, it would be better to write "Focused on Hospital Works" as an example in the title of the paper.

Thank you for this comment. We agree and have changed the title to better reflect the context of the study, it is now titled: “Coping with workplace incivility in hospital teams: How does team mindfulness influence prevention and promotion focused emotional coping”.

-Team Mindfulness is the most important keyword of this study. In Section 1.2, it is necessary to describe or define their definition of Team Mindfulness by the authors of this paper.

Although we provide a definition of team mindfulness on page 2, we have sought to further clarify our understanding of team mindfulness in the revised manuscript. Specifically, we write (p. 2-3):

“Team mindfulness refers to “a shared belief among team members that team interactions are characterized by awareness and attention to present events, and by experiential, non-judgmental processing of within-team experiences” [19] (p. 326). The construct involves two separate components: (1) Present focused attention, which involves “paying attention to what is happening in the moment” and (2) experiential processing, which involves “observing stimuli in an open-minded fashion, without judgment or evaluation” [19] (p. 325-326). In this respect, team mindfulness is a shared perception regarding the extent to which team interactions are present focused and non-evaluative. We argue that an attentive, open-minded, and non-evaluative team approach may buffer the effects of incivility and increase (or decrease) specific coping responses.”

-The questionnaire is written in Chinese. Is it sure that there are no errors in the translation process into English? This is because the questionnaire itself is bound to be somewhat abstract.

The research team is experienced in conducting rigorously designed survey investigation in China. Most scales were well-established and have previously been used in China, however for a few of the more novel scales, we followed translation and back-translation procedures. We had very good communication with the nurses, and they informed us that the survey was quite reader friendly, and they did not encounter any difficulties in filling the survey. Indeed, this is reflected in the fact that the alpha for all scales was above .7.

- On the other hand, describe whether it is confident that the 526 participants in the survey gave very true answers.

It is impossible to determine whether ‘true’ answers were provided by the participants, however it should be noted that the prevalence of incivility was similar to that reported in other studies (i.e. around 90%; Pearson & Porath, 2012). We note the prevalence rate (90.1%) in the manuscript on page 6, and had the participants responded in a more socially desirable manner, we might have expected the prevalence rate of incivility to be lower. Nevertheless, it should be noted that only 157 participants were willing to provide information on the source of the incivility. Therefore, it seemed that the sample generally provided accurate information on the extent to which they encountered incivility, but many did not want to provide information on who enacted the uncivil behavior. This information is now included in the manuscript on page 6:  

Prior to the formal analyses, we explored the data to better understand the nature of the incivility experienced by the respondents. At time 1, 474 respondents (90.1%) had experienced some form of incivility in the past month. However, of these, only 157 (33.1%) were willing to indicate the source of the incivility. These 157 respondents most commonly cited patients’ relatives as the main perpetrators (55.4% of cases), followed by patients (24.2%), nurses (7.6%), managers (6.4%), doctors (4.5%), and consultants (1.9%).        

-Chapter 4. Discussion has very abstract contents, so it is recommended to present the overall argument as an easily understandable picture.

We have provided more specific information on the extent to which the hypotheses were supported in the discussion on page 8. We have also amended the conclusion section to give a clearer overview of the main study findings.

-4.1 Limitations and Future Directions should be combined with Chapter 5. In other words, it is recommended to change the title of Chapter 5 to 5. Conclusions and Future Research Works and describe future research tasks. On the other hand, the limitations of this study described in Section 4.1 are natural due to the nature of this study, so there is no need to describe it.

Thank you for this suggestion, we have changed the title of Chapter 5 to ‘Limitations, future research directions, and conclusions’.

Thank you again for your advice and suggestions, which we believe have helped us to clarify and strengthen the contribution of this work.

Round 2

Reviewer 2 Report

Dear Authors, thank you for incorporating most of my suggestions. The paper looks much improved now. 

Author Response

Many thanks indeed.